# CodeChain: An Open, Million-scale Dataset for Code Language Models at the Repository Level

## Abstract

Code large language models (LLMs) have shown remarkable advances in code understanding and generation tasks. Programming corpora serve as the foundation for various code LLMs. In reality, repositories consist of multiple files with numerous cross-file dependencies. Leveraging the dependency information can effectively enhance the code understanding and generation capabilities. However, existing works fail to utilize dependencies effectively. Consequently, there is a pressing need for an open dataset that specifically focuses on capturing and leveraging the cross-file dependencies. To fill in this gap, we release CODECHAIN, an augmentation of the code dataset at the repository level, provides a rich context for code LLMs to learn from. Specifically, to capture the cross-file dependencies, we first parse the code project into a topological graph where nodes represent files and edges denote dependencies. Then, we employ a novel random walk method to determine the code chain and concatenate the corresponding files. To utilize such corpus for supervised fine-tuning, we design CHAIN-INSTRUCT to enable the model to thoroughly learn the code contents and its dependencies. Ultimately, we produce 562,587 code chains and 1,021,550 instruction samples. With CODECHAIN, we train our model on multi-task learning objectives and evaluate on the public benchmarks. The experimental results demonstrate that model by learning the interconnected nature of codes significantly outperforms the previous methods, showcasing the effectiveness of CODECHAIN in advancing the code understanding and generation.

## 1 Introduction

Code large language models (LLMs) have shown remarkable advances in code understanding, completion, and generation tasks. Code corpora, as the foundation of large code models Li et al. (2023); Guo et al. (2024); Zheng et al. (2023); Rozière et al. (2023); Luo et al. (2023), are from diverse sources: open-source repositories, platforms, forums, and so on. Models pre-trained and finetuned with such data exhibit strong code understanding and analysis capabilities. For repository-level code corpora, previous work decide to directly concatenate code files, which ignores the dependency information between files within the project. To utilize dependency information, DeepSeek-Coder Guo et al. (2024) adopts a topological sorting method, which orders files based on the number of its dependencies of each file. Nonetheless, only considering the the number of dependencies can not reflect the true dependency order between files. Moreover, it struggles to determine the correct order when multiple files have identical dependency counts, leading to errors. More comparison analysis are in Section 4.1.

To fill in this gap, we design a simple yet effective random walk method based on file dependencies to determine the code chain. Specifically, we adopt the approach used by DeepSeek-Coder Guo et al. (2024) to convert code files into a topological graph. In this graph, nodes represent files, and edges symbolize the dependencies between them. The random walk begins by selecting a random starting node on the graph. From there, subsequent nodes are chosen based on their dependency relationships, continuing until the cumulative in-degree of the selected nodes surpasses a predefined threshold. Our method not only captures the correct order of files when multiple nodes have the same in-degree but also maps out indirect call relationships between files.

Utilizing this pipeline, we curate a dataset CODECHAIN. This dataset provides a comprehensive and accurate framework of file dependencies, enhancing the learning process for code LLMs. Additionally, we compile an instruction dataset tailored for supervised fine-tuning, generating 1,021,550 instruction samples. With CODECHAIN, we fine-tune our model and assess its performance on recent well-known benchmarks. The results indicate that our model significantly surpasses previous models in code understanding and generation capabilities by learning the interconnected nature of code files, thereby underscoring the value of CODECHAIN in advancing these areas. The contributions of our paper are as follows:

- **CODECHAIN**. We develop and release a comprehensive data curation pipeline. Ultimately, we compile 562,587 code chains, which can serve as training corpora for code language models, significantly enhancing code understanding and generation capability.

- **Repository Algorithm**. We introduce a straightforward yet effective random walk method for extracting dependencies. Unlike DeepSeek-Coder, our approach reliably generates dependencies in the true file order, which is essential for accurately handling code repositories.

- **CHAIN-INSTRUCT**. Focusing on instruction fune-tuning, we design five distinct tasks for each code chain that vary in length from 2 to 4, which results in the creation of 1,021,550 instruction samples.

- **Impressive Performance**. We assess the performance of our fine-tuned models on well-known benchmarks. The models demonstrate exceptional performance by effectively grasping the intricate interconnections within the repository, markedly outperforming previous models. This highlights the efficacy of CODECHAIN in advancing code understanding and generation capabilities.

## 2 RELATED DATASET WORK

In recent years, numerous large language models (LLMs) have been developed specifically for code-related tasks. Code LLMs Feng et al. (2020); Chen et al. (2021); Scao et al. (2022); Li et al. (2022); Allal et al. (2023); Fried et al. (2022); Wang et al. (2021); Bai et al. (2023); Guo et al. (2024) pre-trained on billions of code snippets from diverse sources (e.g. GitHub), such as Starcoder Li et al. (2023); Lozhkov et al. (2024), CodeLlama Rozière et al. (2023), and DeepSeek-Coder Guo et al. (2024). The development and refinement of Code LLMs have been pivotal in automating software development tasks, providing code suggestions, and supporting natural language to code translations. Besides, there exits several open-source code fine-tuning datasets: Magicoder-OSS-Instruct [1], Python code subset of ShareGPT [2], Magicoder-Evol-Instruct [3], and Evol-Instruct-Code [4]. These instruction datasets primarily focus on enhancing the ability to understand code and solve coding problems within a single file without considering cross-file dependencies. While our CODECHAIN can efficient utilize such dependency information, providing a repository-level understanding.

## 3 DATA CURATION PROCESS

**Overview:** The overall process of data curation is in Figure 1. Our curation mainly consists of seven parts: data crawling and filtering, dependency graph generation, CodeChain generation, quality screening, Chain-Instruct generation, GPT4 Reviewing, and human-in-the-loop testing.

### 3.1 GITHUB DATA CRAWLING AND FILTERING

CODECHAIN is a large-scale dataset consisting of code texts concatenated based on dependency relationships at the repository level. To initiate the data collection process, our first step is to gather repositories. We collect public repositories created before May 2024 on GitHub and retain python language. In order to tackle any potential data leakage concerns, we focus on repositories that have

---

[1] https://huggingface.co/datasets/ise-uiuc/Magicoder-OSS-Instruct-75K
[2] https://huggingface.co/datasets/ajibawa-2023/Python-Code-23k-ShareGPT
[3] https://huggingface.co/datasets/ise-uiuc/Magicoder-Evol-Instruct-110K
[4] https://huggingface.co/datasets/nickrosh/Evol-Instruct-Code-80k-v1

Figure 1: Data Curation Framework. There are seven parts in our framework to gain the high quality repository-level datasets.

been recently created and are not forks. To ensure better data quality, we start the downloading process sequentially from the ones with the highest stars count. Besides, we apply filtering rules similar to Deepseek-Coder (Guo et al., 2024) to preliminarily filter out lower-quality code. By applying these filtering rules, we reduce the total amount of data to only 86.7% of its original size. We briefly describe the filter rules: Firstly, we filter out files with an average line length exceeding 100 characters or a maximum line length surpassing 1000 characters. Additionally, we remove files with fewer than 25% alphabetic characters. After removing the duplicate ones, 55264 repositories are retained. We provide more details on the diverse sources of CODECHAIN in Appendix G.

## 3.2 DEPENDENCY GRAPH GENERATION

After downloading repositories from GitHub, our next step is to get the dependency relationships for each repository. In order to better construct the code-chains from the dependency relationships, we save the dependency relationships in graphs which we call dependency graphs. Each dependency graph is a directed graph that can describe the calling relationships between files within a repository. In this graph, each node represents a code file and it points to the code files that import it. To generate the dependency graphs, we read every code file in the same repository and consider various expressions in code for importing modules to extract the dependency relationships. The expressions included are as follow:

- **Basic import(and rename)**: Use the "import" keyword followed by the module name or "as" keyword to give it an alias, like "import xx","import xx as xxx".
- **Import specific content from a module(and rename)**: Use the "from" keyword followed by the module name and "import" keyword (and "as" keyword), such as "from a import b as c".
- **Import multiple functions**: use a comma-separated list within the import statement for importing multiple functions from a module or package,like "from mymodule import function1, function2".
- **Absolute references**: An absolute reference specifies the complete path to a resource from the root directory. For instance, "from mypackage.mymodule","import myfunction".
- **Relative references**: A relative reference specifies a path starting from the current location in the directory structure. For example, "from . import sibling_module".

Our dependency parser can analyze these expressions to identify the imported modules. It then searches to ascertain whether the referenced modules correspond to files within the same repository. This process starts from the root directory of the current program and adheres to code's import mechanism. The process of our dependency parser is written as the *HasDependency()* function in

Algorithm 1. Then if file A import file B, we add edge from B to A in the dependency graph. It's worth noting that in a repository, the dependency relationships may not necessarily be fully contiguous. Hence, for the same repository, there might not be only one dependency graph.

## 3.3 CODE-CHAIN GENERATION

---

**Algorithm 1** Random Walk for Codechains Generation

---

1: **procedure** CODECHAINGENERATION($files$,$threshold$)
2:     $total\_degree \leftarrow 0$
3:     $selected\_chain \leftarrow [\,]$
4:     $graph \leftarrow \{\}$
5:     $inDegree \leftarrow \{\}$
6:     **for each** $file$ **in** $files$ **do**
7:         $graph[file] \leftarrow [\,]$         ▷ Initialize empty adjacency list for file
8:         $inDegree[file] \leftarrow 0$         ▷ Initialize in-degree as 0 for file
9:     **end for**
10:
11:     **for each** $fileA$ **in** $files$ **do**         ▷ Generate a list of files
12:         that fileA depends on
13:         $dependencies \leftarrow [fileB \text{ for } fileB \text{ in } files \text{ if } \text{HASDEPENDENCY}(fileA, fileB)]$
14:         **for each** $fileB$ **in** $dependencies$ **do**
15:             $graphs[fileB].\text{append}(fileA)$     ▷ If A import B,
16:         Add edge from B to A
17:             $inDegree[fileA] \leftarrow inDegree[fileA] + 1$    ▷ Increment in-degree of A
18:         **end for**
19:     **end for**
20:
21:     **while** $total\_degree < threshold$ **do**
22:         $degree \leftarrow 0$         ▷ Randomly Select a item
23:         as the start of a chain
24:         $initial\_item \leftarrow \text{RANDOMCHOICE}(files)$
25:         $newlist.\text{append}(initial\_item)$
26:         $degree \leftarrow degree + inDegree[initial\_item]$
27:         **while** $random\_item \neq null$ **do**
28:             $random\_item \leftarrow \text{RANDOMCHOICE}(graphs[random\_item])$
29:             $newlist.\text{append}(random\_item)$     ▷ Randomly select the next item in
30:         files depend on this item
31:             $degree \leftarrow degree + inDegree[random\_item]$
32:             **if** $newlist$ not in $selected\_chain$ **then**
33:                 $seletcted\_chains.\text{append}(newlist)$
34:                 $total\_degree \leftarrow total\_degree + degree$
35:             **end if**
36:         **end while**
37:     **end while**
38:     **return** $selected\_chain$
39: **end procedure**

---

We next generate code-chains from the dependency graphs. If we select a path on a dependency graph, the path could be seen as a sequential chain representing a subset of the file dependency relationships within a repository, where each node is imported by the next node it points to. We term a path like this a code-chain. The Algorithm 1 describes the process of code-chains generation on the dependency graphs in one repository. The generation process consists of two steps: **In-degree calculation** and **Random Walk**.

To compute the in-degree for each node, the algorithm initializes with the creation of two data structures: an empty adjacency list dictionary named "graphs" to map dependencies between files, and an empty dictionary named "in_degree" to keep track of the number of dependencies each file has. The algorithm proceeds by iterating over each file to establish dependencies between pairs of

files. If "fileA" depends on "fileB", "fileA" is added to the adjacency list of "fileB" in "graphs", which means creating an directed edge from "fileB" to "fileA". Concurrently, the in-degree of "fileA" is incremented in the "in_degree" dictionary. Then, we set up a threshold to limit the number of chains generated by restricting the cumulative sum of in-degrees of all chain nodes across the entire repository. We build the code-chains from the dependency graphs based on their degrees and through a way of random selections. Specifically, The process begins with the random selection of a node within the graph, which serves as the starting point for initializing a chain. Then, this process continues by randomly selecting the next node from among those that are dependent on the current one, until no further nodes are available. If the cumulative degree stays below a predefined threshold, the newly formed chain is added to a list of selected chains. This process repeats until the total degree of all chains exceeds the threshold. When we gain all the code chains, we remove the duplicate ones.

### 3.4 QUALITY SCREENING AND TEXT CONCATENATION

After generating the code chains, each code file within the same chain is concatenated to form a training sample. A comment indicating the chain between files is added at the beginning of each file to incorporate dependency information. In addition to applying the filtering rules outlined in Section 3.1, we exclude any repository that does not encompass all its files in the chains or lacks dependency relationships to maintain file integrity. Furthermore, to prevent contamination from public benchmarks like HumanEval, we adopt the same n-gram filtering process used by DeepSeek-Coder Guo et al. (2024). Specifically, we remove any code segment from CODECHAIN that contains a 10-gram sequence matching one found in the test data. For test data strings shorter than 10-grams but at least 3-grams long, we use precise matching to ensure exclusion. These quality screening measures reduce the dataset to 61.5% of its original size.

### 3.5 CHAIN-INSTRUCT GENERATION

To optimize the use of dependency information during the fine-tuning process of models, we meticulously design five distinct instruction tasks: **predicting dependencies from code files, completing code based on dependencies, writing README files, creating API documentation, and generating configuration files**. Detailed examples are available in the Appendix K.

We construct the instruction datasets for code-chain that range in length from 2 to 4. Specifically, the tasks of predicting dependencies from code and completing code based on dependencies are automated using scripts. However, the other three tasks cannot be automated. For these, motivated by previous work Wang et al. (2023a); Xu et al. (2023), we employ specific prompts to engage GPT-4 OpenAI (2023) in assisting with their completion. The concrete prompts are in Appendix D.1. Besides, we consider GPT4 (OpenAI, 2023) as a reference and supervisor to reflect on the data quality. Entries that do not meet our stringent quality standards are meticulously flagged and subsequently removed. Please refer to Appendix D.2 for detailed prompts. Ultimately, we gain a total of 1,021,550 unique samples.

### 3.6 HUMAN-IN-THE-LOOP

Due to the volume of data reaching millions, individually inspecting each record requires a significant amount of manpower and resources. To address this, After the GPT4 scanning, we recruit 20 college students specializing in software engineering to conduct manual sample inspections, and calculate the data quality pass rate. Specifically, we first randomly select 10,000 samples from the dataset. To minimize the impact of subjective judgment, we provide all participating students with comprehensive training and guidelines. During the assessment phase, each student evaluates the quality of the samples, deciding whether they are acceptable or unacceptable. To ensure evaluation accuracy, we implement a cross-validation method, ensuring that each sample is reviewed by at least three different students. Furthermore, we establish a consensus mechanism based on the principle that the minority should conform to the majority, resolving any ties or disagreements. The outcome of these inspections consistently shows a 95.6% pass rate, affirming the high quality of our data.

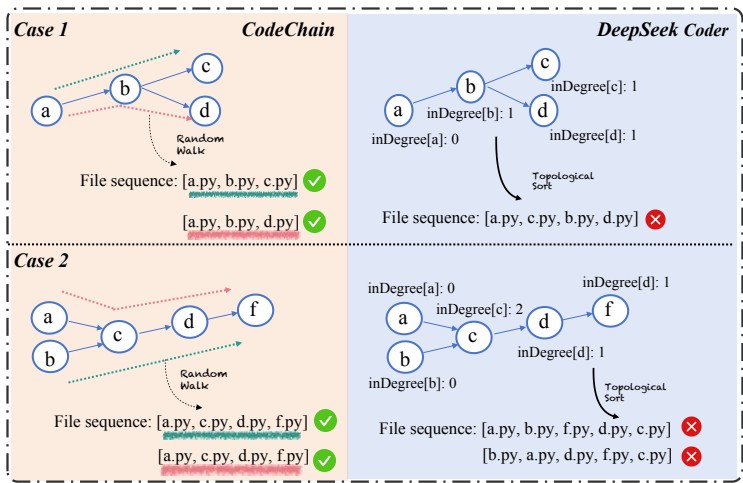

Figure 2: Comparison with DeepSeek-Coder: The image displays two rows, each illustrating different cases where DeepSeek-Coder yields incorrect rankings. The results from our method are highlighted in red, while those from DeepSeek-Coder are shown in blue.

# 4 ANALYSIS OF CODECHAIN

## 4.1 CODECHAIN VS. DEEPSEEK-CODER

In DeepSeek-Coder Guo et al. (2024), the dependencies among files are analyzed on topological graphs, with the in-degree of a node indicating the number of dependencies a file has. The files are then sorted based on in-degrees as input for the model. However, this method does not accurately reflect the actual dependencies in a repository. Figure 2 demonstrates specific errors that occur during the sorting. DeepSeek-Coder's inaccurate file sorting stems from two main issues: 1) The algorithm produces incorrect ordering when multiple nodes share the same in-degree. In Case 1, within a branched structure, it is obvious that two files at a fork do not have any dependency. However, through topological sorting, each file is forcibly assigned an order. Moreover, when encountering multiple nodes with identical minimum in-degrees, the algorithm assigns their order randomly. This leads to an inaccurate file sequencing based on the order of invocation, overlooking the indirect invocation relationships among the files, as evidenced by nodes $d$ and $f$ in case 2.

2) The in-degrees might not accurately reflect the actual file dependencies in the invocation chain. Placing the node with the highest in-degree at the end would lead to incorrect ordering. For example, node $c$ in case 2 has the highest in-degree yet is located in the middle of the chain. To address this, we adopted a random walk strategy to establish an invocation chain based on file dependencies. The method starts by randomly choosing an initial node and then continues by randomly selecting subsequent nodes that are dependent on the current node. This continues until no further nodes are available. The whole iteration ends until the cumulative in-degree of all nodes exceeds a predetermined threshold. This approach ensures that the sequence accurately mirrors the true invocation order and maintains the time complexity at $O(n)$. The proof of the complexity can be seen in Appendix E

## 4.2 STATISTIC FEATURES

In this section, we provide the statistics of CODECHAIN. Specifically, in Figure 3, we show the distribution of the length of code-chains, and the distribution of the lines of code files containing in each code-chain in Figure 4. Please refer to Appendix F for more statistics. these statistical analyses demonstrate the diversity of our dataset and the completeness of the data distribution.

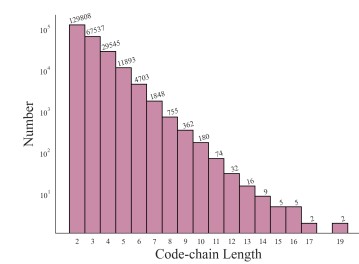 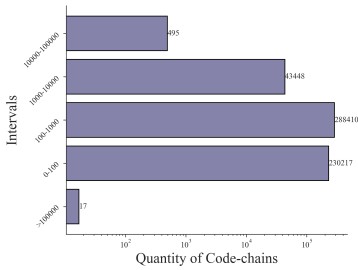

Figure 3: Distribution of Code-chain Length. We count the number of chains of varying lengths.

Figure 4: Distribution of lines of code files in Code-chains. We count the lines of code files containing in each code-chain.

## 5 EVALUATION

### 5.1 EVALUATION BENCHMARK

We evaluate the ability of single-file complement on HumanEval Chen et al. (2021) benchmark and cross-file complement on CrossCodeEval Benchmark. HumanEval is a crafted collection of 164 Python programming problems to test the abilities of code generation models. Cross-file code completion requires the model to access and understand repositories that span multiple files with numerous cross-file dependencies.

### 5.2 EVALUATION METRICS

**Pass@k.** We adopt the Pass@k metric Chen et al. (2021) in HumanEval benchmark. We denote the total number of successfully passing test cases as $k$, thus Pass@k:

$$\text{Pass@k} = \mathbb{E}\left[1 - \frac{\binom{n}{k-c}}{\binom{n}{k}}\right] \tag{1}$$

where $n$ is the total number of generated samples for each problem, and $c$ is the number of correct generated code snippets passing all the test cases ($n > k \geq c$).

**Code Match.** The code match metric evaluates generated code accuracy by comparing it to reference code using Exact Match (EM) and Edit Similarity (ES). These metrics assess the precision of the code completion process, considering elements like identifiers, keywords, and operators.

### 5.3 IMPLETMENTATION DETAILS

Code-Llama and DeepSeek-Coder-Base are used as the base models for supervised fine-tuning (SFT). All experiments are conducted with 16 NVIDIA A100-80GB GPUs. The learning rate first increases into $8 \times 10^{-5}$ with 50 warmup steps and then adopts a cosine decay scheduler. We adopt the Adam optimizer Kingma & Ba (2015) with a global batch size of 64 samples. For HumanEval evaluation, we adopt EvalPlus Liu et al. (2023) for evaluation. For CrossCodeEval Ding et al. (2023), we set the maximum sequence length to 2048 tokens, the maximum output length to 50 tokens, and a limit of 512 tokens for the cross-file context.

### 5.4 EXPERIMENTAL RESULTS

As CODECHAIN consists of only python code, we give the comparison results on python language on benchmarks.

**Single-File Benchmark Evaluation** Table 1 shows that models finetuned with our CHAIN-INSTRUCT significantly beat the base models and recent open-source baselines, closing the gap with GPT-3.5 and GPT-4 in HumanEval benchmark. From Magicoder Wei et al. (2023), Wavecoder Yu

Table 1: Evaluation results of Pass@1 on HumanEval. We use self-reported scores whenever available. All methods use greedy decoding.

| Models | Base Model | Params | Instruction Data | Model Weight | HumanEval |
|---|---|---|---|---|---|
| *Proprietary Models* | | | | | |
| GPT-3.5 Turbo | - | - | - | - | 72.6 |
| GPT-4 Turbo | - | - | - | - | 85.4 |
| *Open-source Models* | | | | | |
| phi-2-2.7B Gunasekar et al. (2023) | - | 2.7 B | ✗ | ✔ | 49.8 |
| CHAIN-INSTRUCT (ours) | phi-2-2.7B | 2.7 B | ✗ | ✔ | 65.2 |
| StarCoder Li et al. (2023) | - | 15B | ✗ | ✔ | 33.6 |
| WizardCoder Luo et al. (2023) | StarCoder | 15B | ✔ | ✔ | 57.3 |
| OctoCoder Muennighoff et al. (2023) | StarCoder | 15B | ✔ | ✔ | 46.2 |
| WaveCoder-SC Muennighoff et al. (2023) | StarCoder | 15B | ✔ | ✔ | 50.5 |
| CodeGeex2 Zheng et al. (2023) | ChatGLM | 6B | ✔ | ✔ | 36 |
| Code-Llama Rozière et al. (2023) | - | 7B | ✗ | ✔ | 33.5 |
| Code-Llama-Instruct Rozière et al. (2023) | Code Llama | 7B | ✔ | ✔ | 34.8 |
| WaveCoder-CL Yu et al. (2023) | Code Llama | 7B | ✔ | ✔ | 48.1 |
| Magicoder-CL Wei et al. (2023) | Code Llama | 7B | ✔ | ✔ | 60.4 |
| CHAIN-INSTRUCT (ours) | Code Llama | 7B | ✔ | ✔ | 65.4 |
| DeepseekCoder Guo et al. (2024) | - | 1.3 B | ✗ | ✔ | 33.9 |
| DeepseekCoder | - | 6.7B | ✗ | ✔ | 49.4 |
| DeepseekCoder | - | 33 B | ✗ | ✔ | 56.1 |
| CHAIN-INSTRUCT (ours) | Deepseek-Coder | 1.3*B* | ✔ | ✔ | 64.7 |
| WaveCoder-DS Yu et al. (2023) | Deepseek-Coder | 6.7B | ✔ | ✔ | 64.0 |
| Magicoder*S*-CL Wei et al. (2023) | Deepseek-Coder | 6.7B | ✔ | ✔ | 70.7 |
| CHAIN-INSTRUCT (ours) | Deepseek-Coder | 6.7B | ✔ | ✔ | 74.3 |
| CHAIN-INSTRUCT (ours) | Deepseek-Coder | 33 B | ✔ | ✔ | 77.4 |

et al. (2023) and CODECHAIN, we can see the effectiveness of instruction datasets from code snippets. Besides, the results also demonstrate that the information of code dependency is effective for understanding and generating code. At the same time, for multilingual evaluation, we conduct more tests on MBPP benchmark Austin et al. (2021) in Appendix I. Besides, we test the effects of dependencies in context on ODEX Wang et al. (2023b) for general API usage. More results are in Appendix J.

**Repo-level Benchmark Evaluation**   For Repo-level Evaluation, we first assess the performance of current open-source models on the CrossCodeEval Ding et al. (2023). The results, displayed in Table 2, reveal that models with CHAIN-INSTRUCT consistently excel over competitors in cross-file completion tasks, highlighting the dependency information enhanced effectiveness in practical applications. Notably, this dataset is established between March and June 2023. In contrast, our CHAIN-INSTRUCT dataset deliberately omits code repositories created during this timeframe. This exclusion guarantees that the dataset is not part of our training data, effectively preventing any data leakage. Besides, we conduct evaluation on RepoBench-C Liu et al. (2024) in Appendix H.

## 5.5   ABLATION AND ANALYSIS

**Effect of Repository Generation Algorithm.**   In the section, we compares our random walk for repo-level code generation with DeepSeek-Coder. To ensure a more equitable evaluation, both the CodeChain and DeepSeek-Coder pipelines are applied to the same dataset (CodeChain's source data). Since DeepSeek-coder doesn't open source the code of their data processing pipline, we reproduce it according to technical report, then we obtain the DeepSeek-Instruction dataset based on the same data source. Comparison results are in Table 4, which further proves the advantages of CHAIN-INSTRUCT. Furthermore, Table 3 demonstrates the ability of our algorithm to effectively capture dependencies. We conduct a statistical analysis to measure the percentage of the graph's edges explored by our algorithm during the chain generation process. This analysis includes all nodes and edges within the dependency graphs. In this context, a unique path in a graph is defined as a sequence of edges that connects a series of nodes (vertices) without revisiting any node. These unique paths represent the various ways in which nodes are interconnected through their dependencies in the graph.

**Effect of Data Quantity**   In Figure 5, both fine-tuned models initially score low but show improvement as more data is introduced, with scores rising from a data ratio of 0.2 to 1.0. While scores continue to increase with more data, the rate of improvement slows at higher data ratios, sug-

Table 2: Performance of different models on cross-file code completion.

| Model | Size | Python | |
|---|---|---|---|
| | | EM | ES |
| *Close-source Models* | | | |
| GPT-3.5-turbo | Unknown | 4.88% | 52.58% |
| GPT-4-Turbo (Nov 2023) | Unknown | 20.66% | 66.92% |
| *Open-source Models* | | | |
| phi-2-2.7B | 2.7B | 5.28% | 55.17% |
| CodeGeex2 | 6B | 8.11% | 59.55% |
| StarCoder-Base | 7B | 6.68% | 59.55% |
| CodeLlama-Base | 7B | 7.32% | 59.66% |
| DeepSeek-Coder | 1.3B | 4.18% | 50.65% |
| DeepSeek-Coder | 6.7B | 9.53% | 61.65% |
| DeepSeek-Coder | 33B | 11.68% | 62.82% |
| *Models Fine-tuned on* CHAIN-INSTRUCT | | | |
| phi-2-2.7B + CHAIN-INSTRUCT | 2.7B | 8.20% | 58.44% |
| CodeLlama-Base + CHAIN-INSTRUCT | 7B | 16.85% | 66.75% |
| DeepSeek-Coder + CHAIN-INSTRUCT | 1.3B | 4.18% | 50.65% |
| DeepSeek-Coder + CHAIN-INSTRUCT | 6.7B | 20.13% | 67.15% |
| DeepSeek-Coder + CHAIN-INSTRUCT | 33B | 22.62% | 68.12% |

Table 3: The percentage of graph edges explored by our algorithm during the process of generating unique dependency chains.

| Total Nodes | Total Edges | Nodes Explored | Edges Explored | Unique Paths Found | Proportion of Nodes Covered | Proportion of Edges Covered |
|---|---|---|---|---|---|---|
| 1110392 | 1508577 | 1045921 | 1453490 | 523563 | 94.2% | 96.4% |

gesting that the models may be nearing their performance peak or that additional data contributes less significantly to further gains.

**Effect of the length of the Code Chain** Given that our instruction chains vary in length from 2 to 4, we allocate an equal amount of data (20,000 samples) for fine-tuning the model across each chain length. In Figure 6, performance improves as code chain length increases for both models. Longer chains significantly boost performance by providing more contextual information, which enhances pattern recognition and the handling of complex code structures. This leads to better predictions and higher quality code generation, allowing models to effectively manage more intricate programming tasks.

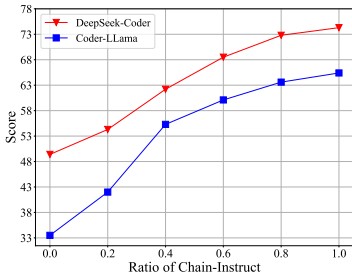

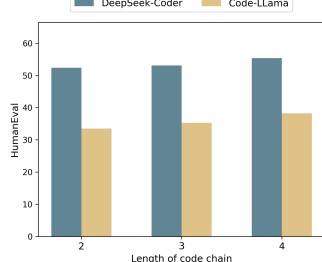

Figure 5: Ablation on Data Quantity. We examine the impact of varying training data ratios (0.2 1.0) on model performance.

Figure 6: Ablation on length of code chain. We give the performance of two fine-tuned models across varying lengths of code chains (2, 3, and 4).

Table 4: Results of CODECHAIN and DeepSeek-Coder on Repository Generation Algorithm.

| Model | Parameter | HumanEval | CCEval (EM) | CCEval (ES) |
|---|---|---|---|---|
| DeepSeekCoder-6.7B-Base | 6.7 B | 49.4 | 9.53% | 61.65% |
| DeepSeekCoder-6.7B-Base + CodeChain-Instruction | 6.7 B | 74.3 | 20.13% | 67.15% |
| DeepseekCoder-6.7B-Base + DeepSeek-Instruction | 6.7 B | 69.7 | 14.25% | 63.33% |

## 6 CONCLUSION

In this paper, we introduce a million-scale dataset, CODECHAIN, designed for training code LLM at the repository level. We employ a novel random walk method to capture cross-file dependencies and concatenate files to form code chains. Additionally, we create an instruction dataset, CHAIN-INSTRUCT, to enhance the model's learning of code contents and dependencies. Extensive evaluations on public benchmarks confirm CODECHAIN's effectiveness in code understanding and generation.

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

## A    LIMITATIONS

CodeChain focuses on some popular language repositories, potentially excluding insights from other programming languages. Also we need to set a better stopping strategy for ramdom walk in the future instead of using a threshold. Moreover, the criteria used for quality screening may inadvertently favor certain types of code or repositories, potentially excluding valuable but unconventional coding practices from the dataset.

## B    SOCIAL IMPACT OF DATASET

**Potential benefits**: CodeChain provides a richer context for LLMs to learn from by augmenting the code pre-training dataset at the repository level. This richer context can lead to improved model performance and understanding of code semantics. We release a comprehensive data curation pipeline for anyone to use to get more different code language repositories. Our method ensures the generation of dependencies in the true file order, which is crucial for handling code repositories. This could help others to build upon our works and furthur advance the ability of Code LLMs at the repository level.

**Potential risks**: Collecting code from over 50,000 repositories from GitHub raises potential privacy and legal concerns, especially if the code includes proprietary or copyrighted material without proper authorization or consent. Like any large-scale dataset sourced from online repositories, CodeChain may inadvertently capture and perpetuate biases present in the original data sources. This could lead to biased model predictions and reinforce existing societal inequalities in code development and usage.

## C    CROWDSOURCING

In conducting our study, we identified several potential risks to participants. Firstly, there is a risk to privacy and confidentiality, as participants are required to share personal information. To mitigate this, all data will be anonymized and stored securely, with access restricted to authorized personnel only. Secondly, there may be psychological risks, such as discomfort or stress during the tasks. To address this, we have included detailed instructions and debriefing sessions to ensure participants feel supported throughout the process. Additionally, participants have the right to withdraw from the study at any time without penalty. Lastly, while there are no significant physical risks associated with our procedures, we will monitor participants for any signs of distress and provide appropriate support. We pay each participant an hourly rate of $10. The primary participants we recruit are college students.

## D    PROMPT TEMPLATES

### D.1    PROMPTS FOR CHAIN-INSTRUCT GENERATION

---

**Generate README File**

Please generate a comprehensive README document for the project, utilizing the provided file contents and outlined dependency relationships.

The README may include the following sections: a clear project title, a brief description of the project's purpose, installation instructions, usage guidelines, a list of dependencies with explanations of their roles, code examples where applicable, and a section on how to contribute to the project.

Ensure that the documentation is user-friendly, technically accurate, and formatted for easy readability. Include any necessary warnings or notes that users must be aware of when interacting with the project.

---

---

**Generate Interface Documentation**

Please generate detailed interface documentation for the software project, using the provided code file contents and dependency relationships.

The documentation may comprehensively describe each function and class in the code. Include the following details for each interface component: a clear description, parameters with types and descriptions, return values and their types, exceptions that might be thrown, and example usage scenarios.

Ensure that the documentation is well-structured, maintaining a consistent format across sections. The document should also include an introduction to the interface, its overall purpose, and any specific considerations or compatibility issues. Aim for clarity, accuracy, and utility to aid developers in understanding and implementing the interfaces effectively.

---

**Generate Configuration File**

Please create a comprehensive configuration file for the project, using the provided file contents and detailed dependency relationships.

The configuration file may include key-value pairs or settings that are essential for the operation of the project. Ensure to include sections categorizing different types of settings such as database connections, API keys, environment variables, and other critical infrastructure components. Each entry should be clearly commented to explain its purpose, expected values, and any dependencies it has on other settings. Format the file for easy navigation and modification, using consistent indentation and spacing. Also, provide guidelines at the beginning of the file on how to correctly update or modify these settings to meet specific deployment environments or use cases.

---

## D.2    PROMPT FOR DATA QUALITY

The prompt we employ for GPT-4 is deliberately tailored to elicit insightful evaluations and is as follows:

---

**Quality Prompt**: You are now a data grader. You will grade the data I provide according to my requirements, explain the reasons, and then give a piece of higher-quality data based on this piece of data.

Please help me rate the following dialogue data and explain the reasons. Require:

1. Scoring perspective: whether the problem belongs to the field of code; whether the description is clear; whether the answer is accurate; whether the language is coherent;

2. Point scale: 5-point scale, 1 point: very poor; 2 points: slightly poor; 3 points: barely qualified; 4 points: usable; 5 points: excellent.

3. Please rate the answer. If the score is lower than 5 points and higher than 2 points, a higher quality data will be generated based on this piece of data.

4. Format: You can only return a parsable json format data, no other content. For example: "score": 4, "reason": "", "modified-data": "". Among them, score represents the score for this question, reason represents the reason for the score, and states the advantages and disadvantages of the data, and modified-data represents You generated a new, higher-quality data based on the above data. Compared with the data provided, this new data solves the shortcomings you mentioned above and is directly available.

5. All reasons are written in reason.

6. If the score is lower than 5 points and higher than 2 points, modified-data must be provided.

7. Modified-data contains a complete piece of data that is directly available, and the quality must be higher and more in line with the quality of ChatGPT's training data. If null needs to be output, replace it with None. Now please follow the above requirements to annotate the following conversation data and return your annotated results in pure json form: "".

---

## E    PROOF OF TIME COMPLEXITY

Let:

- $n$: The total number of files in the repository.

- $T(n)$: The time complexity of the algorithm.

- $degree(x)$: The degree or in-degree of item $x$ (which represents its connections or dependencies).

### E.1 OUTER LOOP

The outer loop runs while the condition $total\_degree < threshold$ is satisfied. Initially, the degree is set to 0. A random item is chosen from the file list, denoted by:

$$\text{Initial\_item} = \text{RandomChoice(files)}$$

This selected item is added to a new list:

$$\text{newlist} \leftarrow \text{newlist} \cup \{\text{Initial\_item}\}$$

The degree is updated by adding the in-degree of the selected item:

$$\text{degree} \leftarrow \text{degree} + \text{inDegree(Initial\_item)}$$

Each of these operations—random selection, list update, and degree update—has a constant time complexity of $O(1)$.

### E.2 INNER LOOP

When a random item exists (i.e., $random\_item \neq \varnothing$), the inner loop is executed. A new random item is selected from the graph:

$$\text{random\_item} = \text{RandomChoice}(graphs[\text{random\_item}])$$

This item is then added to the list:

$$\text{newlist} \leftarrow \text{newlist} \cup \{\text{random\_item}\}$$

The degree is updated once again by adding the in-degree of the newly selected item:

$$\text{degree} \leftarrow \text{degree} + \text{inDegree(random\_item)}$$

Before appending the updated list to the set of selected chains, the algorithm ensures that this list is not already present:

$$\text{if newlist} \notin \text{selected\_chains, then append newlist to selected\_chains}$$

Finally, the total degree is incremented by the newly computed degree:

$$\text{total\_degree} \leftarrow \text{total\_degree} + \text{degree}$$

### E.3 TIME COMPLEXITY ANALYSIS

The operations in the inner loop, like random selection, list append, and degree updates, all have constant time complexity $O(1)$. The only exception is the check to see whether $newlist \notin selected\_chains$, which, in the worst case, has a time complexity of $O(n)$, where $n$ is the number of files, since it involves searching the list.

The outer `while` loop runs while $total\_degree < threshold$. Since the total degree accumulates based on the file selections, the number of iterations of the outer loop is proportional to $n$. Therefore, the overall time complexity can be expressed as:

$$T(n) = \sum_{1}^{k} c$$

where $k$ represents the number of iterations of the outer loop, and $c$ is the constant time per iteration. Given that $k$ is proportional to $n$, the total time complexity simplifies to:

$$T(n) = O(n \cdot c) = O(n)$$

Thus, the algorithm has a time complexity of $O(n)$.

## F STATISTIC DETAILS

Here, we give more statistic details for CODECHAIN in Table 5. We trace back the word frequency across all repository titles to better understand the contents of these repositories. The word cloud Figure 7 shows the top 30 words with the highest frequency.

Table 5: Statistic details of our CODECHAIN

| Features | Statistics |
|---|---|
| Size | 8.65GB |
| The Number of Chains | 562587 |
| The Number of Repos | 31182 |
| Average Chain Length | 1.79 |
| The Number of Chains (chain length $>1$) | 246776 |
| Average Chain Length (chain length $>1$) | 2.81 |
| Number of CHAIN-INSTRUCT | 1,021,550 |

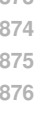
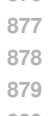
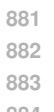
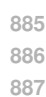
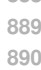

Figure 7: Word Cloud of CODECHAIN.

## G    SOURCE OF CODECHAIN

In CODECHAIN, the diversity of data sources is essential for ensuring the broad applicability our datasets. We categorize the collected repositories into four classes based on their functionalities: Artificial Intelligence, Web Development, Data Processing, and Others. Figure 8 shows the distribution of repositories with different functionalities, with Artificial Intelligence and Web Development collectively accounting for over half of the total. Specifically, Computer Vision technologies lead at 21.2%, followed by Natural Language Processing at 8.6%, and Machine Learning at 5.1%. In Web Development, Frontend and Backend technologies collectively represent 17.6% and 9.1% of our data sources respectively. Additionally, Network programming account for 10.3% of the total distribution. Within Data Processing, the subcategories include Crawlers, Data Analysis, and Automated Scripts. The "Other" category includes miscellaneous items that do not fit into the main categories, which include Tutorials, Plugins, and other miscellaneous items.

## H    MORE RESULTS ON REPOBENCH

In this section, we conduct experiments on RepoBench-c, more results are in Table 6. CHAIN-INSTRUCT performs better than other baselines.

## I    SCALABILITY TO OTHER LANGUAGES

Our framework for generating code chains can be easily extended to support various programming languages. Based on the same data processing pipeline, we collect and generate fine-tuning datasets for the Java and C++ languages as a case to show the scalability of our pipelines. At the same time, based on this multilingual version, we conduct more tests on MBPP benchmark Austin et al. (2021).

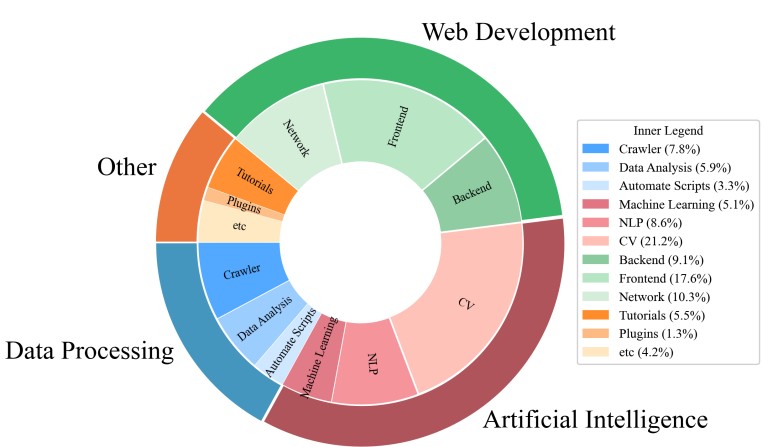

Figure 8: Sources of CODECHAIN.

Table 6: the results on the RepoBench-C. The results show that CHAIN-INSTRUCT helps improve the generation and understanding capabilities at the repo-level.

| Model | Size | ALL | |
|---|---|---|---|
| | | EM | ES |
| CodeGen | 350M | 32.44 | 71.08 |
| CodeGen | 2.7B | 27.35 | 68.30 |
| CodeGen | 6.1B | 31.67 | 70.68 |
| StarCoder | 15.5B | 31.67 | 71.28 |
| Codex | 175B | 31.31 | 72.22 |
| DeepSeekCoder-6.7B-Base | 6.7B | 32.44 | 71.08 |
| DeepSeekCoder-6.7B-Base + CHAIN-INSTRUCT | 6.7B | 34.12 | 72.42 |
| DeepseekCoder-33B-Base | 33B | 34.02 | 72.15 |
| DeepSeekCoder-33B-Base + CHAIN-INSTRUCT | 33B | 35.83 | 73.03 |

## J  MORE RESULTS ON ODEX

More results on ODEX(General API usage) are in Table 8.

## K  MORE EXAMPLES OF CHAIN-INSTRUCT

We provide five cases for five instructions. As you can see in Figure 9, Figure 10, Figure 11, Figure 12, and Figure 13,

Table 7: Results on MBPP.

| Model | Parameter | MBPP |
|---|---|---|
| DeepSeek-Coder-base | 1.3 B | 47.7 |
| DeepSeek-Coder-base + CHAIN-INSTRUCT | 1.3 B | 55.0 |
| DeepSeek-Coder-base | 6.7 B | 58.7 |
| DeepSeek-Coder-base + CHAIN-INSTRUCT | 6.7 B | 65.3 |
| DeepSeek-Coder-base | 33 B | 65.3 |
| DeepSeek-Coder-base + CHAIN-INSTRUCT | 33 B | 69.3 |

Table 8: Results on ODEX

| Model | Parameter | pass@1 | pass@2 | pass@5 |
|---|---|---|---|---|
| DeepSeek-Coder-base | $6.7B$ | 41.28 | 46.12 | 53.47 |
| DeepSeek-Coder-base + CodeChain-Instruction | $6.7B$ | 45.36 | 51.07 | 58.10 |
| DeepSeek-Coder-base | 33B | 49.28 | 58.55 | 66.17 |
| DeepSeek-Coder-base + CodeChain-Instruction | 33B | 51.87 | 61.24 | 70.25 |

```
'Instruction': "Analyze the following code snippet to determine the dependency relationship between files."
"input": {

'Chatbot/chatbotconfig.py':
"""
import os
basedir = os.path.abspath(os.path.dirname(__file__))
class Config(object):
    SECRET_KEY=os.environ.get('SECRET_KEY') or 'you-will-never-guess'
""" ,
'Chatbot/chatbot/__init__.py':
"""
import flask
from flask import Flask
from chatbotconfig import Config
app=Flask(__name__)
app.config.from_object(Config)

import keras
import nltk
import pickle
import json
from keras.models import load_model
from nltk.stem import WordNetLemmatizer
lemmatizer=WordNetLemmatizer()

model=load_model('chatbot_codes/mymodel.h5')
intents = json.loads(open('chatbot_codes/intents.json').read())
words = pickle.load(open('chatbot_codes/words.pkl','rb'))
classes = pickle.load(open('chatbot_codes/classes.pkl','rb'))
from chatbot import routes
""" ,
'Chatbot/chatbot.py':
"""
from chatbot import app
"""
},
"output": ['Chatbot/chatbotconfig.py', 'Chatbot/chatbot/__init__.py', 'Chatbot/chatbot.py']
```

Figure 9: Case 1 of instruction tasks. The task is to analyze the following code snippet to determine the dependency relationship between files.

```
{
"Instruction": "Complete the corresponding code content based on the provided file contents and dependency relationships.",
"input":
{
"dependency":  ['Chatbot/chatbotconfig.py', 'Chatbot/chatbot/__init__.py', 'Chatbot/chatbot.py'],

'Chatbot/chatbotconfig.py':
"""
import os
basedir = os.path.abspath(os.path.dirname(__file__))
class Config(object):
    SECRET_KEY=os.environ.get('SECRET_KEY') or 'you-will-never-guess'
""",
'Chatbot/chatbot/__init__.py':
"""
import flask
from flask import Flask
from #"The content to be completed."

app=Flask(__name__)
app.config.from_object(Config)
import keras,nltk
import pickle
import json
from keras.models import load_model
from nltk.stem import WordNetLemmatizer
lemmatizer=WordNetLemmatizer()
model=load_model('chatbot_codes/mymodel.h5')
intents = json.loads(open('chatbot_codes/intents.json').read())
words = pickle.load(open('chatbot_codes/words.pkl','rb'))
classes = pickle.load(open('chatbot_codes/classes.pkl','rb'))
from chatbot import routes
""",
'Chatbot/chatbot.py':
"""
from chatbot import app
"""
},
"output":
"""import flask
from flask import Flask
from chatbotconfig import Config
"""
}
```

Figure 10: Case 2 of instruction tasks. The task is to complete the corresponding code content based on the provided file contents and dependency relationships.

```
{
"Instruction": "Generate a README document for the project based on the following file contents and dependency relationships.",
"input": {
"dependency": ['Chatbot/chatbotconfig.py', 'Chatbot/chatbot/__init__.py', 'Chatbot/chatbot.py'],

'Chatbot/chatbotconfig.py':
"""
import os
basedir = os.path.abspath(os.path.dirname(__file__))
class Config(object):
    SECRET_KEY=os.environ.get('SECRET_KEY') or 'you-will-never-guess'

""" ,
'Chatbot/chatbot/__init__.py':
"""
import flask
from flask import Flask
from chatbotconfig import Config

app=Flask(__name__)
app.config.from_object(Config)

import keras,nltk
import pickle
import json
from keras.models import load_model
from nltk.stem import WordNetLemmatizer
lemmatizer=WordNetLemmatizer()

model=load_model('chatbot_codes/mymodel.h5')
intents = json.loads(open('chatbot_codes/intents.json').read())
words = pickle.load(open('chatbot_codes/words.pkl','rb'))
classes = pickle.load(open('chatbot_codes/classes.pkl','rb'))
from chatbot import routes
""" ,
'Chatbot/chatbot.py':
"""
from chatbot import app
"""

},
"output": {
"README.md":
"""
# Project Overview
## File Dependency Relationships
- `chatbotconfig.py` depends on `chatbot/chatbot/__init__.py`
- `chatbot/chatbot/__init__.py` depends on `chatbot/chatbot.py`

## File Descriptions
- `chatbot/chatbotconfig.py`: Contains configuration settings for the chatbot.
- `chatbot/chatbot/__init__.py`: Initializes the chatbot application and loads necessary dependencies such as Flask, Keras, NLTK, and trained mode
- `chatbot/chatbot.py`: Entry point for the chatbot application.
"""
}
```

Figure 11: Case 3 of instruction tasks. The task is to generate a README document for the project based on the following file contents and dependency relationships.

```
{"Instruction": "Generate interface documentation based on the following code file contents.",
"input": {
'Chatbot/chatbotconfig.py':
"""
import os
basedir = os.path.abspath(os.path.dirname(__file__))
class Config(object):
    SECRET_KEY=os.environ.get('SECRET_KEY') or 'you-will-never-guess'
""" ,
'Chatbot/chatbot/__init__.py':
"""
import flask
from flask import Flask
from chatbotconfig import Config
app=Flask(__name__)
app.config.from_object(Config)
import keras, nltk. pickle, json
from keras.models import load_model
from nltk.stem import WordNetLemmatizer
lemmatizer=WordNetLemmatizer()
model=load_model('chatbot_codes/mymodel.h5')
intents = json.loads(open('chatbot_codes/intents.json').read())
words = pickle.load(open('chatbot_codes/words.pkl','rb'))
classes = pickle.load(open('chatbot_codes/classes.pkl','rb')
from chatbot import routes
""" ,
'Chatbot/chatbot.py':
"""
from chatbot import app
"""
},
"output": {
"Interface Documentation": {
## File: chatbotconfig.py
### Functions and Classes:
- `Config`: Class containing configuration settings for the chatbot.
- `SECRET_KEY`: A secret key for the application, retrieved from the environment variables or set to a default value if not found.
---
## File: chatbot/__init__.py

### Dependencies:
- `Flask`: Web framework for creating the chatbot application.
- `chatbotconfig.Config`: Configuration settings imported from `chatbotconfig.py`.

### Loaded Dependencies:
- `keras`: Deep learning library for building and training models.
- `nltk`: Natural Language Toolkit for natural language processing tasks.
- `pickle`: Python module for serializing and deserializing objects.
- `json`: Python module for working with JSON data.
- `load_model`: Function for loading a Keras model from a file.
- `WordNetLemmatizer`: Class for lemmatizing words in text data.

### Variables:
- `app`: Flask application instance.
- `model`: Loaded Keras model for the chatbot.
- `intents`: JSON data representing intents for the chatbot.
- `words`: Pickled data containing processed words.
- `classes`: Pickled data containing class labels.

### Routes:
- `routes`: Module containing routes for the chatbot application.
---

## File: chatbot.py

### Dependencies:
- `app`: Flask application instance imported from the `chatbot` package.
}}}
```

Figure 12: Case 4 of instruction tasks. The task is to generate interface documentation based on the following code file contents.

```
{"Instruction": "Write a configuration file for the project based on the following file contents and their dependency relationships.",
"input": {
"dependency": ['Chatbot/chatbotconfig.py', 'Chatbot/chatbot/__init__.py', 'Chatbot/chatbot.py'],

'Chatbot/chatbotconfig.py':
"""
import os
basedir = os.path.abspath(os.path.dirname(__file__))
class Config(object):
    SECRET_KEY=os.environ.get('SECRET_KEY') or 'you-will-never-guess'

""",
'Chatbot/chatbot/__init__.py':
"""
import flask
from flask import Flask
from chatbotconfig import Config
app=Flask(__name__)
app.config.from_object(Config)

import keras, nltk, pickle, json
from keras.models import load_model
from nltk.stem import WordNetLemmatizer
lemmatizer=WordNetLemmatizer()
model=load_model('chatbot_codes/mymodel.h5')
intents = json.loads(open('chatbot_codes/intents.json').read())
words = pickle.load(open('chatbot_codes/words.pkl','rb'))
classes = pickle.load(open('chatbot_codes/classes.pkl','rb'))
from chatbot import routes
""",
'Chatbot/chatbot.py':
"""
from chatbot import app
"""
},
"output": {
"Project configuration file": """
[dependencies]
Chatbot/chatbotconfig.py = [Chatbot/chatbot/__init__.py]
Chatbot/chatbot/__init__.py = [Chatbot/chatbot.py]
Chatbot/chatbot.py = []

# Other configuration items
[dependencies]
Chatbot/chatbotconfig.py = [os]
Chatbot/chatbot/__init__.py = [flask]
Chatbot/chatbot.py = []

"""}}
```

Figure 13: Case 5 of instruction tasks. The task is to write a configuration file for the project based on the following file contents and their dependency relationships.

