# OpenReview forum: "CodeChain: An Open, Million-scale Dataset for Code Language Models at the Repository Level"
_ICLR.cc/2025/Conference — ICLR 2025 Conference Withdrawn Submission_

### Official Review · Reviewer_x4YZ · 2024-10-25

**Soundness:** 2
**Presentation:** 1
**Contribution:** 2
**Rating:** 3
**Confidence:** 4

**Summary:**

The paper introduces CodeChain, a large-scale dataset designed to improve the performance of code language models by capturing cross-file dependencies in software repositories. Traditional code models often fail to leverage these dependencies, leading to suboptimal understanding and generation of code. CodeChain addresses this by using a random walk algorithm to generate code chains from repository-level data, creating over 562,000 code chains. CHAIN-INSTRUCT, a set of instructions for fine-tuning, is also introduced to train models on tasks such as dependency prediction and code completion. The evaluation shows that models trained on CodeChain outperform previous methods on various benchmarks, enhancing code understanding and generation across single-file and multi-file projects.

**Strengths:**

1. The author studied a critical task, constructing a pre-training dataset for repository-level code completion and generation tasks widely present in real-world scenarios.
2. By utilizing the dependencies between files, the author merged file contents into a single context based on the random walk algorithm, constructing the CodeChain dataset while also creating an instruction fine-tuning dataset called CHAIN-INSTRUCT.
3. By fine-tuning the model on the constructed dataset and demonstrating the effectiveness of the proposed dataset on a wide range of downstream benchmark datasets (including single-file and multi-file benchmarks).

**Weaknesses:**

1. The method proposed in the paper significantly overlaps the previous research work, deepseek-coder, and is merely an incremental improvement. I believe it mainly replaces the topological sorting technique used in the deepseek-coder with a random walk-based technique. Moreover, I believe that using the topological sorting technique has considerable rationality, as it ensures that the dependent files of the current file are included in the context. The author needs to conduct a more in-depth analysis and explain the advantages of using the random walk algorithm.
2. The author has explored the construction of context using the random walk algorithm very little, merely using the random walk technique to obtain the dependency relationships between the final files, but has not sufficiently considered the selection of the initial nodes for the random walk and the probability choices for jumping between nodes.
3. The writing style of the article needs improvement; the introduction section requires a more detailed explanation of the motivation and an introduction to related work. The related work section should also include relevant work on constructing pre-training data [1, 2]. Additionally, I suggest that the author adjust the organization of the experimental section, as the data construction method proposed by the author mainly targets cross-file level completion and generation, so the main experimental results in the text should focus on and explain the cross-file benchmarks.
[1]Ding, Hantian, et al. "Fewer Truncations Improve Language Modeling." Forty-first International Conference on Machine Learning.
[2]Shi, Weijia, et al. "In-Context Pretraining: Language Modeling Beyond Document Boundaries." The Twelfth International Conference on Learning Representations.

**Questions:**

Please refer to Weaknesses section.

---

### Official Review · Reviewer_tqjK · 2024-10-28

**Soundness:** 2
**Presentation:** 2
**Contribution:** 2
**Rating:** 5
**Confidence:** 4

**Summary:**

This paper investigates an instruction-tuning strategy for code LLMs to utilize context-file dependencies. The authors construct a dataset, CodeChain, and design five instruction tuning tasks to provide dependency information for LLMs to learn. The experimental results demonstrate that the effectiveness of this fine-tuning, which could improve base LLM performance.

**Strengths:**

1. This paper proposes a curated dataset CodeChain that captures the correct dependencies order of files in repositories.
2. Five designed instruction-tuning tasks are introduced to improve Code LLMs with repository-level understanding.

**Weaknesses:**

1. The random walk algorithm terminates when the in-degree of nodes surpasses a predefined threshold. What is the specific value of threshold? How is it selected? Will this threshold influence the performance after fine-tuning?
2. The authors did not mention the rationale of selecting code chains with the length from 2 to 4. As suggested by Section 5.5, if longer chains could significantly improve performance, why not consider choosing longer chains in the instruct-tuning?
3. Why didn't the authors compare to models after instruction-tuning, e.g., DeepSeek-Coder-Instruct, to show the effectiveness of Chain-Instruct?
4. The authors claim that "CodeChain could efficiently utilize cross-file dependencies". However, the evaluation on repository-level benchmark lacks some key details. E.g., Is there any RAG approach used? What is the experimental setting for it?
5. The performance of fine-tuned DeepSeek-Coder-1.3B is missing in Table 2.

**Questions:**

1. What the purpose of using random walk to generate code chains?
2. How did the authors design the five tasks for instruction tuning? What are the goals of these tasks? How could you demonstrate the necessity of each task?
3. If only part of code chains are selected, will contextual information of repository be lost? Could authors provide the proportion of nodes and edges covered by the selected instruction chains?

---

### Official Review · Reviewer_JKYM · 2024-11-01

**Soundness:** 1
**Presentation:** 2
**Contribution:** 2
**Rating:** 1
**Confidence:** 4

**Summary:**

This work revisits the task of training language models on code data with a focus on capturing the file dependencies inside a project. The main goal of this work is to fix the sorting of files inside a repository to correctly account for parent-child dependencies. For this they propose a random walk algorithm to arrange the files into a sequence for autoregressive model training. They develop instruction tasks and train a Chain-Instruct model and evaluate it on the standard code benchmarks.

**Strengths:**

This work is focused on the important problem of capturing the file dependences for project-aware code generation. They develop a preprocessing a comprehensive pipeline of data collection and model training. They design five instruction tasks for code generation. They demonstrate good quality of the pipeline on single-file and repository based coding tasks.
The main strength of this work is the proposed instruction finetuning approach, which seems to result in substantial improvements in many cases.

**Weaknesses:**

Strong weakness: incorrect presentation of the baseline approach and hence false main contribution. It seems that the authors are incorrectly interpreting the topological sorting algorithm from Guo et al. 2024:
- The main contribution of this paper is in "fixing" the ordering of files from a repository graph: they introduce a random walk ordering to replace topological sorting from Guo et al. 2024 (DeepSeek-Coder). In line 300, it is written that in Guo et al. 2024 "the files are sorted based on in-degrees as input for the model", however, this is not exactly true: in line 25 of Algorithm 1 from Guo et al. 2024 the degree of the child is being decreased after adding a parent, so it it incorrect to call this procedure "sorting by in-degree".
- The "incorrect" orderings of DeepSeek-Coder are presented in the Figure 2 (right) claimed to be from topological sort, but the correct topological orderings for the first example (Figure 2) would be [a.py, b.py, c.py, d.py] or [a.py, b.py, d.py, c.py]; and either [a.py, b.py, c.py, d.py, f.py], [b.py, a.py, c.py, d.py, f.py] are for the second example (Figure 2).

Other weaknesses:
- In the Table 1, it seems that the Chain-Instruct model is only compared to the pretrained DeepSeek-Coder model, and not to a stronger DeepSeek-Coder-Instruct.
- only one programming language is considered: Python

References:
- Guo et al. 2024, DeepSeek-Coder: When the Large Language Model Meets
Programming - The Rise of Code Intelligence: https://arxiv.org/pdf/2401.14196

**Questions:**

- Have you observed in practice any incorrect orderings for the approach of Guo et al. 2024, or the comparison in Figure 2 is based on the theoretical analysis?
- Could you add the discussion about the comparison to the DeepSeekCoder-Instruct and the corresponding numbers from their work (for HumanEval dataset)?
- It seems that the overall pipeline between DeepSeekCoder and your work is similar, it would be nicer to highlight the differences between these two pipelines (data collection and preprocessing, training, evaluation).
-  human-in-the-loop part: what are the instructions for the students? what is the consensus mechanism?
- how Exact Match and Edit Similarity metric are computed in line 359?
- Table 2: do you have any intuition why DeepSeek-Coder + Chain-Instruct does not add upon DeepSeek-Coder for 1.3B model?

---

### Official Review · Reviewer_DzJn · 2024-11-03

**Soundness:** 3
**Presentation:** 2
**Contribution:** 3
**Rating:** 6
**Confidence:** 3

**Summary:**

The authors introduce a million-scale dataset, CODE-CHAIN, designed for training code LLM at the repository level. The authors also create a novel random walk method to capture cross-file dependencies and concatenate files to form code chains that outperform similar state-of-the-art algorithms. The authors also release an instruction dataset version of CODE-CHAIN, namely CHAIN-INSTRUCT. Extensive evaluations on public benchmarks confirm both datasets' effectiveness in code understanding and generation.

**Strengths:**

- The work is novel, as few code datasets exist at the repository level.
- The work is original as it outperforms similar datasets and presents a new algorithm to build cross-file dependency code datasets.
- The construction of CODE-CHAIN corpora is clear and follows a high-quality process, which keeps humans in the loop.
- The evaluation of CODE-CHAIN and CODE-INSTRUCT for code-related tasks is clear enough.
- The work presents significant contributions to the field: a knowledge dataset, an instruct dataset, and the novel random walk method.

**Weaknesses:**

### Important Omissions
- The paper does not discuss limitations and future work, which makes it look incomplete. A couple of topics that you could address in both sections are additional programming languages and diversity of instructions.
- In the contributions list, it is unclear if Code-Chain is the knowledge dataset or the curation pipeline. Please explicitly state whether Code-Chain refers to the dataset itself or the pipeline used to create it. This would help readers better understand the key contributions.
- line 261: 'this, After the GPT4 scanning', please elaborate on what specific process is performed with GPT-4.
- Code-Instruct. You have nearly duplicated instructions, as you are repeatedly using the same prompt templates to generate responses, and simply updating the code being passed as context. State-of-the-art instruction tuning practices recommend including diverse samples with different styles that match your evaluation paradigm. Please see 'Questions' for additional details on this point.
- Code-Instruct. No evaluation for the documentation tasks. Please see 'Questions' for additional details.

### Presentation
- line 070: fune-tuning - please correct the spelling.
- Figure 2: The first sequence in case 2 (green underline) seems incorrect. Shouldn't it be b->c->d->f?
- Table 2: Please explain 'EM' and 'ES' respective meanings in the description.

**Questions:**

### Questions
1. How would you address the lack of diversity in instructions? some options I can think of are using multiple prompt templates for each category or using an LLM to rephrase or revise the instructions (i.e., https://arxiv.org/pdf/2311.13246). I recommend addressing the lack of instruction diversity as a limitation of the instruction dataset and discussing an approach to increase instruction diversity in the future.
2. Did you perform an evaluation of documentation tasks for CODE-INSTRUCT? and if not, why are these tasks omitted from the evaluation?

### Suggestions
3. I would not list "impressive performance" as a contribution, simply mention it in the conclusion section.
4. Please make sure to do an additional proofreading round for the whole paper, including images and tables.
5. Table 1: I suggest showing in bold the highest score on HumanEval for each group.
6. Figure 2: The background highlighting for your approach (left side) does not look red. Besides, using colors to reference parts of the image is not helpful for blind-color people. I recommend changing the figure description.

---

### Official Review · Reviewer_F1Qw · 2024-11-03

**Soundness:** 2
**Presentation:** 3
**Contribution:** 2
**Rating:** 5
**Confidence:** 4

**Summary:**

This paper introduces CodeChain, a large-scale code dataset specifically curated for enhancing repository-level code language models. CodeChain compiles data from a wide range of real-world GitHub repositories and incorporates cross-file dependencies by employing a random walk strategy on dependency graphs. Additionally, Chain-Instruct, a unique instructional set, is designed to help models grasp both code content and dependency relationships more comprehensively. Evaluation results on well-known benchmarks highlight that models trained with CodeChain significantly outperform existing methods.

**Strengths:**

1.	CodeChain offers an extensive dataset that captures dependencies between files, a critical feature for repository-level code understanding and generation.

2.	Through the Chain-Instruct tasks, the dataset encourages models to be dependency-aware, significantly benefiting the generation of contextually correct code across files.

3.	The use of a random walk algorithm to create meaningful dependency chains is innovative, offering a practical solution to preserve file relationships accurately. The results show that random walk leads to better results than previous methods.

**Weaknesses:**

1.	CodeChain’s dependency information is limited to the file level, which may not be as useful for fine-grained dependency analysis (e.g., function calls), thus potentially limiting the model’s practical applications.

2.	The single-file evaluation was conducted on the HumanEval dataset, which does not fully represent the complexity of real-world repositories. Using more complex, real-world benchmarks, such as CrossCodeEval, would enhance the evaluation’s relevance.

3.	The comparison in Table 7 shows Chain-Instruct fine-tuned models outperforming DeepSeek-Coder-base; however, since DeepSeek-Coder-base has not gone through any supervised-finetuning, I think the comparison here is unfair. A more rigorous comparison with a fine-tuned version of DeepSeek-Coder would provide a solid conclusion of CodeChain’s advantages.

4.	At present, the dataset is not available online, which limits reproducibility and external validation.

5.	CodeChain currently contains only Python code, which restricts its applicability to multi-language code understanding or generation tasks.

**Questions:**

1.	Have the authors considered applying additional metrics, such as CodeBLEU, to evaluate model performance?

2.	Why not evaluate the single-file completion task on the CrossCodeEval as well?

3.	Are there plans to extend the dataset to include other programming languages?

---

### Note · Authors · 2024-12-02

I have read and agree with the venue's withdrawal policy on behalf of myself and my co-authors.